# Unique Ultrasonographic Findings of Isolated IgG4-Related Lymphadenopathy

**DOI:** 10.3390/diagnostics11122213

**Published:** 2021-11-27

**Authors:** Jae Sung Yun, Seoyun Choi, Kyu Yun Jang, Eun Hae Park

**Affiliations:** 1Department of Radiology, Ajou University School of Medicine, Suwon 16449, Korea; taurii22@gmail.com; 2Musculoskeletal Imaging Laboratory, Ajou University Medical Center, Suwon 16449, Korea; 3Department of Radiology, Jeonbuk National University Medical School, Jeonju 54907, Korea; 234657@gmail.com; 4Department of Pathology, Jeonbuk National University Medical School, Jeonju 54907, Korea; kyjang@jbnu.ac.kr; 5Research Institute of Clinical Medicine of Jeonbuk National University-Biomedical Research Institute of Jeonbuk National University Hospital, Jeonju 54907, Korea

**Keywords:** IgG4-related disease, lymphadenopathy, ultrasound

## Abstract

IgG4-related disease is a rare immune-mediated disease that can involve many organs in the body. The lymph node is also where IgG4-related diseases occur, but its histological structure is different from that of other organs. For this reason, pathologists have difficulty diagnosing IgG4-related lymphadenopathy. If there were specific imaging findings of IgG4-related lymphadenopathy, it would be of great help to pathologists. A 64-year-old male visited our hospital with right ankle pain. On physical examination, the right lower extremity showed severe swelling with wound dehiscence, and infection was suspected. On CT (128-MDCT, Somatom Definition Flash, Siemens Healthcare) taken at the lower extremity, multiple enlarged lymph nodes were incidentally noted in the right inguinal area. On ultrasonography, a “starry night sign” resembling hyperechoic follicles was observed in the enlarged lymph node. A core needle biopsy was performed, and IgG4-related lymphadenopathy was diagnosed. Laboratory examination showed hypergammaglobulinemia with marked elevated serum IgG4, corresponding to IgG4-related disease. Chest and abdominal imaging were evaluated, but there was no extranodal IgG4-related disease. IgG4-related lymphadenopathy showed a very unique ultrasonography imaging finding. The cortex was filled with diffusely scattered hyperechoic foci and some bright foci gathered to form a follicle. This imaging finding may help diagnose IgG4-related lymphadenopathy.

## 1. Introduction

Immunoglobulin G4 (IgG4)-related disease is a recently discovered, rare immune-mediated condition with infiltration by abundant IgG4-positive plasma cells [1]. The pancreas is most often involved, but symptoms can occur in many organs of the body [2,3]. For example, IgG4-related diseases frequently occur in the lymph node. In imaging studies, regional lymph node involvement is detected adjacent to the affected organ in 80% of patients with IgG4-related disease [4]. However, lymphadenopathy as an initial manifestation of IgG4-related disease is rare [2,3,5,6].

While extranodal IgG4-related diseases characteristically show specific pathologic findings such as dense lymphoplasmacytic infiltration, storiform fibrosis, and obliterative phlebitis in a hematoxylin and eosin stain [1,5,6,7], IgG4-related lymphadenopathy may not show one or more of them, resulting in pathologic confirmation challenging.

For evaluation of lymphadenopathy, ultrasound (US) is the modality of choice to evaluate superficial regions including the head and neck, axilla, and inguinal area [8,9,10]. It would be helpful to both clinicians and pathologists if US could show characteristic findings of IgG4-related lymphadenopathy. However, image findings, including US, of IgG4-related lymphadenopathy are not well known so far. Only Wu et al. [11] reported on US imaging of an IgG4-related disease that occurred in the form of cervical lymphadenopathy.

In the present case, we describe computed tomography (CT), magnetic resonance (MR), and characteristic US imaging findings of a patient with right inguinal and popliteal lymph nodes as the only affected site for an IgG4-related disease.

## 2. Case Report

Our study was approved by our Institutional Review Board, and informed consent was waived. A 64-year-old male came to the emergency room at our institution for right ankle pain. He was diagnosed with psoriasis 22 years ago and had a history of 4 surgeries on the right ankle for fracture and septic arthritis within the last 3 years. On physical examination, the right lower extremity showed diffuse and severe swelling with redness and heating sensation, and there was wound dehiscence at both the medial and lateral ankle. The range of motion of the right ankle was near full, and motor and sensory abilities were normal.

Laboratory examination showed a normal range of white blood cell counts (5.51 k/µL) and eosinophils (0.19 k/µL). C-reactive protein (42.74 mg/L) and erythrocyte sedimentation rate (77 mm/hr) were elevated, suggesting inflammation. In addition, hemoglobin level (11.0 g/dL) and creatinine (0.44 mg/dL) were low.

The patient underwent right lower extremity CT for the evaluation of cellulitis. Chronic osteomyelitis with acute aggravation was suspected in the distal tibia and calcaneus with diffuse subcutaneous swelling extending from the foot to the distal thigh. In addition, there were multiple enlarged round-shaped lymph nodes in the right external iliac and inguinal areas with a shorter diameter of 0.8–2.1 cm (Figure 1). Because it was too large to be reactive lymph nodes from cellulitis, a biopsy was recommended. For the biopsy, US was performed using an US unit (Philips, EPIQ 7G) and a 5–18 MHz broadband linear array transducer in the right inguinal area. US demonstrated enlarged lymph nodes with echogenic hilum and thickening of the hypoechoic cortex. The unique finding was that the cortex was filled with diffusely scattered hyperechoic foci, and some bright foci gathered to form a follicle, which ended up looking like a “starry night sign” (Figure 2). On the color Doppler images, the hilum showed high vascularity, but the cortex follicles did not show blood flow. An US-guided core needle biopsy was followed by targeting the follicles of the cortex in the enlarged lymph nodes.

Histology showed follicular hyperplasia with many plasma cells (Figure 3). Immunohistochemically, many plasma cells were IgG4-positive (>100 per high power field). CD20 was positive on B cells, and CD10, BCL6, and Ki67 were positive on germinal center cells. BCL2 was negative on germinal center cells. CD3 and CD5 were positive on T cells. The cells were negative for both EBER expression and human herpesvirus 8 (HHV-8). No neutrophil polymorph accumulation or necrosis was demonstrated to suggest active reactive lymph node enlargement. Additional laboratory examination showed hypergammaglobulinemia (2212.8 mg/dL) with marked elevated serum IgG4 (2500 mg/dL). The final diagnosis was IgG4-related lymphadenopathy.

Consequentially, chest CT, abdominal CT, and pelvic MR (axial T1-weighted image, axial T2 weighted image, axial fat suppressed T2 weighted image, 3T Achieva, Philips, Netherlands) were taken to look for other affected organs. There were no organs around the external iliac and inguinal chain to suspect an IgG4-related disease. Size of pancreas was normal, and there was no evidence of involvement in the hepatobiliary tract, kidneys, retroperitoneum, salivary glands, or lung parenchyma. On MRI, the cortex was very thick in the right inguinal and external iliac lymph nodes, and the signal was decreased on T2WI compared to the lymph nodes on the contralateral side (Figure 4). No follicles were seen inside the cortex of the lymph nodes.

The patient underwent below-knee amputation for ankle osteomyelitis and cellulitis. Systemic corticosteroid therapy was recommended to the patient after controlling the infection of the lower leg, but the patient refused treatment, and follow-up was lost.

## 3. Discussion

In this case, the patient showed multiple enlarged lymph nodes of the unilateral inguinal and external iliac chain. In ultrasonography for inguinal lymph nodes, a characteristic appearance of a “starry night sign” was seen. A biopsy was performed on the inguinal lymph node, and immunohistochemical staining revealed many IgG4-positive plasma cells. On hematologic examination, serum IgG4 was markedly increased.

IgG4-related diseases are immune-mediated fibroinflammatory diseases that may cause irreversible organ damage, resulting in life-threatening events. IgG4-related diseases can involve nearly all organ systems, including the pancreas, hepatobiliary tract, lacrimal glands, salivary glands, retroperitoneum, kidneys, thyroid, muscle, and lymph node [2,3,12]. Considering that non-IgG4-related lymphadenopathy or even nonspecific reactive lymphadenopathies may show a significant increase in IgG4-positive cells [13,14,15,16] and the present patient had an infection at the ipsilateral lower extremity without a sign of extranodal IgG4-related disease involvement, we had to be very careful in diagnosing IgG4-related disease. The diagnostic criteria for IgG4-related disease are as follows [2,11,17]: (1) clinical examination showing characteristic diffuse/localized swelling or masses in one or more organs; (2) hematologic examination showing elevated serum IgG4 concentrations (over 135 mg/dL); and (3) histopathologic examination showing marked lymphoplasmacytic infiltration and storiform fibrosis, as well as organ infiltration by IgG4-positive plasma cells. If all three criteria are met, it is considered definite; if the first and third criteria are met, it is considered probable; and if the first and second criteria are met, it is considered possible. Since there were markedly increased IgG4-positive plasma cells in germinal centers and interfollicular regions on the pathological tissue (>100 per high power field) and all other conditions were satisfied, a definite IgG4-related disease was diagnosed in the present case.

The lymph node is an organ frequently involved in IgG4-related disease [4]. Lymphadenopathy in IgG4-related disease usually occurs in two forms [5,6,11]. First, it occurs in the form of localized or generalized lymphadenopathy associated with an extranodal lesion. Second, in rare cases and in the present case, localized or generalized lymphadenopathy occurs as the initial manifestation of IgG4-related disease without extranodal disease. In this case, differential diagnosis included lymphoma, Castleman disease, nodal metastasis, and other autoimmune diseases, and a biopsy should be performed for further evaluation. Misdiagnosis of IgG4-related disease as Castleman disease and vice versa is possible because Castleman disease also may demonstrate elevation of serum IgG4-level and infiltration of IgG4-positive plasma cells. However, Castleman disease-related lymphadenopathy often involves multiple regions (multicentric Castleman disease) and demonstrates lymphoid follicles with a collapsed germinal center and expansion of mantle area showing an onion skin appearance [6,18,19]. None of the mentioned features were noted in the present case. In addition, HHV-8 was negative, which is known to be frequently positive in Castleman disease.

A proper radiologic feature of IgG4-related lymphadenopathy may be helpful because it is not easy to diagnose IgG4-related lymphadenopathy. The characteristic morphologic features of an IgG4-related disease are known to be as follows: dense lymphoplasmacytic infiltration, storiform fibrosis, and obliterative phlebitis [1,5,6,7]. However, in IgG4-related lymphadenopathy, obliterative phlebitis and storiform fibrosis are usually absent, making it difficult to recognize the IgG4-related disease [1,5,6,7]. As a result, a proper radiologic feature of IgG4-related lymphadenopathy is crucial for the diagnosis. Unfortunately, to date, the radiologic findings of IgG4-related lymphadenopathy are not well known yet. There is only one English report in 2019 on US findings of IgG4-related lymphadenopathy to the best of our knowledge [11]. This report described US findings of cervical IgG4-related lymphadenopathy as multiple matted, ovoid, homogeneous, hypoechoic, and enlarged lymph nodes. However, because further US imaging findings have not been described in this report and the provided picture from the report is not focused on the lymph node itself, there is a limitation as to whether the affected lymph node had hyperechoic foci to call a starry night sign. In our case, ultrasonography showed a unique and interesting appearance of IgG4-related inguinal lymphadenopathy that resembled a starry night. The starry night sonographic appearance is constituted by multiple hyperechoic foci on a hypoechoic background. While the underlying conditions are quite different, there are other reported cases showing a similar imaging finding in the liver in cases of acute hepatitis, lymphoma, leukemic infiltration, and Burkitt lymphoma, along with cases involving the mediastinal tuberculous lymph nodes [20].

The “starry night sign” in the present case is thought to be reminiscent of a follicle visible inside the lymph nodes on pathological tissue. However, since we performed a core needle biopsy rather than resection, there was a limitation on matching pathological tissues and US imaging findings directly, and further evaluation with a larger cohort is required. CT and MR imaging only demonstrated multiple enlarged lymph nodes with a round/oval shape, fatty hilum, and cortical thickening without the internal follicles mentioned above.

In conclusion, we present a rare case of IgG4-related disease with involvement localized only to a lymph node, which showed a unique ultrasound finding of a starry night sign. When a starry night sign is seen in an enlarged lymph node, IgG4-related lymphadenopathy could be considered, and this imaging finding may help in diagnosing IgG4-related lymphadenopathy.

## Figures and Tables

**Figure 1 diagnostics-11-02213-f001:**
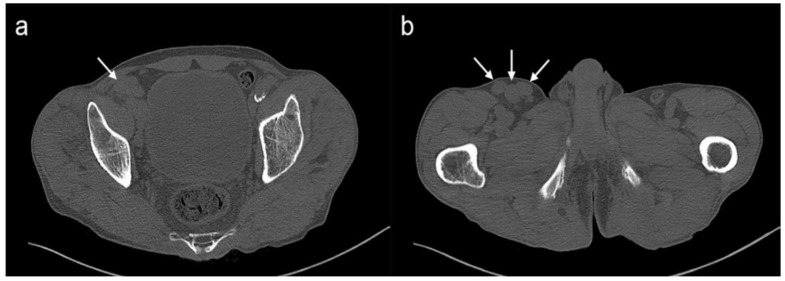
CT images of the lower extremity. Multiple lymphadenopathies (arrows) are seen in (**a**) right external iliac and (**b**) right inguinal areas.

**Figure 2 diagnostics-11-02213-f002:**
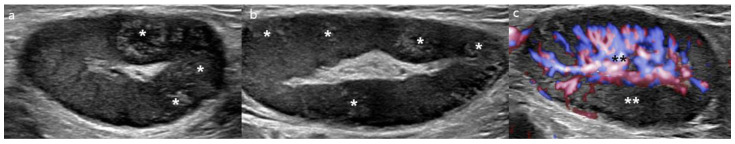
Ultrasonography of the right inguinal lymph node. (**a**,**b**) There are multiple scattered hyperechoic foci in the enlarged hypoechoic cortex. Note some bright foci form the follicle (white asterisks), showing a “starry night sign” with (**a**) short axis and (**b**) long-axis views. (**c**) In the color Doppler image, high vascularity is shown mainly in the hilum (black double asterisks) compared with the cortex (white double asterisks).

**Figure 3 diagnostics-11-02213-f003:**
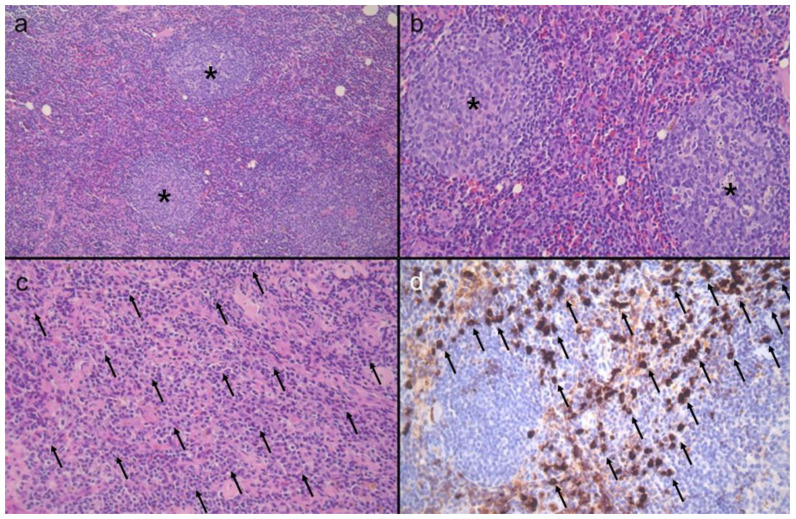
Histologic findings of IgG4-related disease of the lymph node. (**a**) At a low-power image, the lymph node shows follicular hyperplasia accompanying expansion of germinal centers (asterisks). (**b**,**c**) At high-power images, the inter-follicular area between (**b**) secondary lymphoid follicles (asterisks) and (**c**) the paracortex is rich in plasma cells (arrows). (**d**) Immunohistochemically, there are many IgG4-positive plasma cells (arrows). Original magnification: (**a**) ×200 (**b**–**d**) ×400.

**Figure 4 diagnostics-11-02213-f004:**
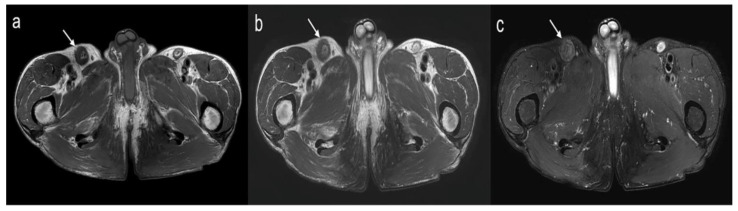
MR findings of the lymph node of IgG4-related disease in the right inguinal and external iliac chain. (**a**) An axial T1-weight image shows right inguinal lymphadenopathy (arrow) with cortical thickening and preserved fatty hilum. (**b**,**c**) Axial T2-weighted Dixon in-phase image (**b**) and axial T2-weighted Dixon water-only image (**c**) show the lower signal of the thickened cortex (arrow) compared to the normal left inguinal lymph node.

## Data Availability

All data generated or analyzed during this study are included in this published article.

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
