# Peer review of "Unique Ultrasonographic Findings of Isolated IgG4-Related Lymphadenopathy"

_diagnostics, 2021, doi:10.3390/diagnostics11122213_

Round 1

Reviewer 1 Report

I have read and evaluated the revised manuscript. The authors have quite satisfactorily addressed my concerns. Therefore, as far as I am concerned, the revised version of the manuscript is quite acceptable.

Reviewer 2 Report

This is an interesting case report representing imaging specifics with the potential to aid clinicians diagnose IgG-related lymphadenopathy.

It seems that the paper has been reviewed and revised accordingly. The work is well presented and the language quality adequate.

My only concern is that one may not fully follow the descriptions provided in the legends of Figures 2 and 3, since no signs have been provided on the images. I recommend the authors to use arrows and/or other easily-identifiable signs on these figures to bring the specific pathologic and imaging changes to the attention of readers.

I recommend its publication after addressing these concerns.

Author Response

This manuscript is a resubmission of an earlier submission. The following is a list of the peer review reports and author responses from that submission.

Round 1

Reviewer 1 Report

Thanks for choosing Diagnostics journal to publish your findings. Please address the following questions. 

1) Just curious, would ultra sound image characteristic findings in inguinal and popliteal lymph nodes as shown in the paper be helpful to diagnose other lymphadenopathy cases ?  To follow up that question, how is this imaging findings different from the one and only reported cervical lymphadenopathy ultrasound imaging from Wu et al.

2) Please check for some minor grammar issues. For example, use “an” instead of “a” in front of Ultra sound word

3) Figure 2 is very crucial to the paper. While it’s understandable we may not get better resolution pictures. But it is necessary to be clear from a readers perspective. So, I would suggest adding arrows in the figure to show those follicles so that readers can have better understanding. Also, in the Doppler image (Figure 2) please indicate with an arrow to show the hilum and cortex region.

4) It would be nice to add a small section of how the samples were made for each testing/examination (MR, CT, US etc).

Reviewer 2 Report

The authors described a case of IgG4-related lymphadenopathy. Only a few minor comments:

  1. Figure 2 legend mentions arrows, which are missing in the figure.
  2. The authors should briefly mention how IgG4 causes inflammation, and why it is localized in most cases.

Reviewer 3 Report

The present case report shows US findings of lymphadenopathy in patients with IgG4-related  disease as diffusely scattered hyperechoic foci and gathered bright foci. The authors claim this findings help diagnose IgG4-related lymphadenopathy.

The reviewer has several concern about their presentation and discussion.

  1. US findings should be correlated with pathologic findings for the purpose of understanding pathophysiological explanation of the US findings as the authors express “starry night sign”.
  2. Starry night sign or starry sky sign is noted in a variety of diseases, such as hepatitis, lymphoma, tuberculous lymphadenopathy. Consideration of the difference of this sign of IgG4-RD and others is helpful to understand the “starry night sign” in IgG4-RD.
  3. This case has complication of severe infection and inflammation in the ipsilateral extremity of the right inguinal lymph nodes. Accompanied immune reaction to inflammation might have major influence on pathologic findings. Please discuss these pathophysiological points.
  4. In case of Castleman disease, elevated IgG4, CRP and platelet count, anemia are common findings similar to the present case. Pathologic finding of follicular formation and IgG 4-positive plasma cells are also mutual findings. Differential diagnosis with Castleman disease is inevitable.
  5. Changes in the LN findings after amputation of the right leg would help making difference of Castleman disease.
  6. Figure 1 should be replaced by CT images with soft tissue condition.
  7. Figure 2 is missing arrows.